# ICP-MS Assessment of Essential and Toxic Trace Elements in Foodstuffs with Different Geographic Origins Available in Romanian Supermarkets

**DOI:** 10.3390/molecules26237081

**Published:** 2021-11-23

**Authors:** Cezara Voica, Constantin Nechita, Andreea Maria Iordache, Carmen Roba, Ramona Zgavarogea, Roxana Elena Ionete

**Affiliations:** 1National Institute for Research and Development of Isotopic and Molecular Technologies, 67-103 Donat Str., 400293 Cluj-Napoca, Romania; cezara.voica@itim-cj.ro; 2National Research and Development Institute for Forestry “Marin Drăcea”—INCDS, 725100 Câmpulung Moldovenesc, Romania; 3National Research and Development Institute of Cryogenics and Isotopic Technologies—ICSI Rm. Valcea, 4 Uzinei Str., 240050 Rm. Valcea, Romania; roxana.ionete@icsi.ro; 4Faculty of Environmental Science and Engineering, Babeş-Bolyai University, 30 Fantanele Str., 400294 Cluj-Napoca, Romania; carmen.roba@ubbcluj.ro

**Keywords:** aromatic spices, trace elements, ICP-MS technique, exposure assessment

## Abstract

The present study was conducted to quantify the daily intake and target hazard quotient of four essential elements, namely, chromium, cobalt, nickel, and copper, and four toxic trace elements, mercury, cadmium, lead, and arsenic. Thirty food items were assigned to five food categories (seeds, leaves, powders, beans, and fruits) and analyzed using inductively coupled plasma-mass spectrometry. Factor analysis after principal component extraction revealed common metal patterns in all foodstuffs, and using hierarchical cluster analysis, an association map was created to illustrate their similarity. The results indicate that the internationally recommended dietary allowance was exceeded for Cu and Cr in 27 and 29 foodstuffs, respectively. According to the tolerable upper level for Ni and Cu, everyday consumption of these elements through repeated consumption of seeds (fennel, opium poppy, and cannabis) and fruits (almond) can have adverse health effects. Moreover, a robust correlation between Cu and As (*p* < 0.001) was established when all samples were analyzed. Principal component analysis (PCA) demonstrated an association between Pb, As, Co, and Ni in one group and Cr, Cu, Hg, and Cd in a second group, comprising 56.85% of the total variance. For all elements investigated, the cancer risk index was within safe limits, highlighting that lifetime consumption does not increase the risk of carcinogens.

## 1. Introduction

Trace elements (TE) bioaccumulation in plants is often associated with food supply chains. Currently, globalization raises significant concerns about reducing contaminants in the environmental ecosystem to avoid substantial health issues. The use of aromatic spices is markedly increasing in most regions of the world [1]. However, only 2% of the world’s herbs and spices are produced in Europe, even though the primary consumer market for spices is represented by Europe and Asia [2]. Additionally, the popularity of aromatic products in the European market is rising, and a recently growing demand for spices has been observed around the world [3]. In addition to culinary purposes to improve the color, aroma, palatability, and acceptability of food, spices are used in folk medicine as antiscorbutic, antispasmodic, tonic, and carminative agents to treat bronchitis ulcers, and as diuretics, depuratives, and vermifuges. Various spices are used for their antioxidant [4], anti-inflammatory [5], and antimicrobial [6] action in the treatment of diabetes [7] and even in cancer therapy [8]. Spices contain nutrients and bioactive compounds that positively influence organism functioning, such as vitamins C and E, carotenoids, and phenolic compounds (flavonoids, tannins, and flavones), and may play a beneficial role in reducing the risk of cardiovascular diseases [9].

Aromatic plant nutritional and medicinal properties may be interlinked through both nutritional and non-nutritional phytochemicals. The biochemical processes that regulate the main stages of growth in the herb that are further used as spices depend significantly on the natural nutrient intake from the soil. The elemental content of aromatic spices depends on their biological and geographic origins. The TE sample profile represents a natural proxy for natural and anthropogenic activities [10]. Thus, industrial development, combined with environmental pollution, increases toxic elements in spices and herbs [11]. Pb, Cd, Hg, Ni, Co, Cr, and As have been reported to have hepatotoxic, nephrotoxic, neurotoxic, or immunotoxic effects on human health and have been linked to severe diseases such as Alzheimer’s or Parkinson’s disease [12]. These elements are potentially hazardous if the maximum daily intake concentrations are exceeded for sustained periods. In considering extreme health risks for humans, the Joint FAO and World Health Organization Expert Committee on Food Additives and European Union Regulation No. 1881/2006 indicate maximum thresholds for Pb (0.3 mg/kg) and Cd (0.2 mg/kg) in plant leaves and fresh herbs [13]. The maximum permissible limits for TE in herbal medicines established by the World Health Organization [14] are 5, 10, 0.3, 2.0, and 0.2 mg/kg for As, Pb, Cd, Cr, and Hg, respectively. According to the European Food Safety Authority (EFSA), preventing and controlling Pb contamination is essential, considering its side effects on the human body and the significant amounts available in the environment [15].

Contamination with TE is by far not the only issue in consuming vegetables. The contamination of soil, groundwater, and surface water with nitrate and nitrite is responsible for plant enrichment with nitrate and nitrite. Due to its natural form and additive presence, a new challenge in the risk-benefit health assessment of vegetable food enriched with nitrate is frequently discussed. In agricultural practices, to prevent the growth of the bacterium *Clostridium botulinum*, impressive amounts of nitrate and nitrite are also used to promote some food colors. Nitrate-accumulating vegetable content varies between species and genotypes with different ploidies, and high contents have been mentioned for *Brassicaceae*, *Chenopodiaceae*, *Amaranthaceae*, and *Apiaceae*. Past studies have indicated that contaminated fennel led to nitrate toxicosis that was associated with cattle side effects (muscular tremors, respiratory distress, and convulsions) [16]. A carcinogenic risk is related to the formation of nitrosamine compounds.

Various techniques have been developed to assess trace element contamination in aromatic spices from plants. Inductively coupled plasma with both mass spectrometric and chemometric methods are powerful analytical techniques used in elemental analysis. Thus, inductively coupled plasma-mass spectrometry (ICP-MS) was used to investigate the mineral composition and geographic origin of Italian saffron, ginger, cinnamon, green cardamom, turmeric, coriander, cumin, curry, and chili [17]. Inductively coupled plasma-atomic emission spectrometry (ICP-AES) and inductively coupled plasma-optical emission spectroscopy (ICP-OES) techniques were appropriate for elemental determination in paprika, cumin, saffron, coriander, anise, mustard, and black pepper [18]. Novel approaches are expected to be used for vanilla and saffron due to the high cost of the raw materials and very complicated conditions for harvest and production [19]. For example, laser ablation-inductively coupled plasma-time of flight mass spectrometry (LA-ICP-MS-TOF) was used to identify the elemental composition of vanilla and determine geographic origin by using discriminant function analysis [20]. Monitoring saffron adulteration is a crucial issue due to the high world market price. Among other methods, ultrasound-assisted extraction has been used in combination with total reflection X-ray fluorescence (XRF) for multi-elemental analysis of various spices and herbs [21].

Romanian cuisine involves significant amounts of food items, and a lack of knowledge regarding the TE content of foodstuffs available in supermarkets requires investigation. Furthermore, monitoring their toxicity would help determine the health impacts of frequently consumed spices and provide essential data on herbs produced in Romania. Therefore, the objective of the present study is (i) to quantify the levels of essential trace elements (Cr, Co, Ni, and Cu) and toxic trace elements (Hg, Cd, Pb, and As) in the food items widely used and commonly available on the market and consumed in the main cities of Romania, (ii) to establish the relationship between trace elements in foodstuff, and (iii) to assess the contribution of essential and toxic trace elements to human nutritional assimilation and the health risk due to the ingestion according to limitation established by WHO/FAO regulation.

## 2. Results and Discussion

### 2.1. Validation of Analytical Methods and Summary Statistics

Each sample was measured in six replicates and compared with the certified reference materials (CRM-NCS ZC85006, CRM-BOVM1, and CRM-IAEA359) to evaluate the accuracy of the method for spice analysis. Recovery percentages in a range of 84–105% (CRM-NCS ZC85006), 86–96% (CRM-BOVM1), and 85–107% (CRM-IAEA359) confirmed no significant loss or gain for each analyte during the digestion procedure. In addition, the agreement between the certified values and the measured values was good, demonstrating the satisfactory performance of the developed method. The correlation coefficient was calculated from calibration curves of each analyte element, and a value of *r* = 0.9999 was obtained. The limit of detection (LOD) was in the range of 0.0005–0.006 (μg/g) (Table 1). The coefficient of variation values ranged between 1.1% and 7.2% (less than the threshold of 8%). The results for the eight elements investigated in a variety of spices were found to be in good agreement with the certified results.

### 2.2. Concentration of Minor Elements in Different Spices

Plants concentrate minerals from the environment that are essential for growth and development, as well as metals with detrimental consequences for living organisms. Frieden [22] divided metals into micro, trace-, and ultratrace elements based on the amounts found in tissues. The World Health Organization [23] classification contains essential elements, probably essential elements, and potentially toxic elements. Trace elements are minerals vital for the human body in amounts of 1 to 100 mg/day or less than 0.01% of total body weight for adults. Ultratrace minerals are required in amounts of less than 0.001 mg/day. The minor elements investigated in the present survey decrease in mean value (± coefficient of variation) in the following order: Cu (5.88 ± 1.08 μg/g), Ni (1.63 ± 0.68 μg/g), Cr (1.29 ± 0.74 μg/g), and Co (0.20 ± 1.29 μg/g). The mean amount of Cu varied between 0.11 μg/g (#20) and 23.95 μg/g (#30), with an interquartile range (q3 − q1) of 4.28 μg/g. The concentration was higher than 10 μg/g in fruits (#30) and seeds (#4 and #6) (Figure 1). Copper, which is vital for metabolism, is required in the human diet since it exhibits various biological functions in enzymatic and redox systems [24]. Copper deficiencies in living organisms are directly linked to physiological problems affecting the nervous system, skeleton, optic nerve function, or bone marrow hematopoiesis. The daily intake of Cu exceeding the limit causes lipid peroxidation in liver homogenates [25].

In the present study, we measured both minimum and maximum values of Ni in leaf samples, ranging from 0.01 μg/g (#16) to 4.28 μg/g (#14) with an interquartile range of 1.42 μg/g. Concentrations (in μg/g) of 3.71 (#2), 3.38 (#22), 2.89 (#5), 2.68 (#24), and 2.53 (#28) were also calculated in seeds, powders, beans, and fruits (Figure 1). Nickel is present in several oxidative states, the most common being Ni^2+^, which is usually associated with oxygen and sulfur as oxides and sulfides, respectively. This element is an essential nutrient involved in morphological functions, growth, and development, including germination, and is naturally present in plant organs. It is frequently associated with primitive systems, such as those in anaerobic bacteria [26]. Although it is not essential for humans, Ni has been studied for its potentially harmful effects on health [27]. Previously reported side effects on human health include toxicity in the respiratory tract, allergy, contact dermatitis, asthma, cardiovascular and kidney diseases, different types of cancer, and significant negative effects in destroying mitochondrial DNA. This element enters the ecosystem from anthropogenic activities (industrialization, mining, waste, fertilization, or combustion of fossil fuels) and natural sources (wind-blown dust, volcanoes, forest fires, etc.). Important sources of nickel in the human body are dietary exposure [28] and tobacco consumption. Each cigarette contains 2.32 to 4.20 μg/g, mainly in the very hazardous form of nickel carbonyl [29]. Modern medicine uses nanotechnology as a vector for drug delivery, such as nickel nanoparticles (NiNPs) [30].

In the last decade, chromium has gained popularity as a nutritional supplement and a component of many multivitamin/mineral products, fortified food, and energy drinks [31]. It is an essential mineral that appears to have beneficial effects in regulating insulin action and improving carbohydrate and lipid metabolism [32]. Our study found values between 0.02 and 4.42 μg/g (q3 − q1 = 1.04). High amounts of 4.42 μg/g (#22), 2.60 μg/g (#23), 2.71 μg/g (#7), and 2.03 μg/g (#11) were measured in powders, seeds, and leaves (Figure 1). Cobalt is a vital trace element found in the liver, kidneys, pancreas, heart, skeletal muscle, and bones, and the majority of cobalt in living organisms is present in the structure of vitamin B_12_ [33]. The values of Co obtained in this study (0.03 μg/g, #20; 0.13 μg/g, #2; 0.00 μg/g, #26; and 0.18 μg/g, #10) were lower than those reported for the same spices in Turkey [34]. In contrast, the Ni contents in #22 and #21 from India were comparable to those reported in the present study [35]. We noted similar amounts of metals for #24 in the case of Cr and Cu [36]. The Cu concentrations in #20, #2, #18, and #26 found in Turkey were higher than those in the present study. Even so, other studies reported lower values in the case of #18 and approximately comparable values for #2 and #26 (Table 2).

The recommended dietary allowance (RDA) for Cu and Cr (0.7–0.9 and 0.015–0.03 mg/day, respectively), tolerable upper level (TUL) for Cu and Ni (10 and 1 mg/day, respectively, in the case of adults aged 30–70 years), and provisional tolerable weekly intake (PTWI) for Pb, As and Cd (7, 15, and 25 μg/kg bw, respectively) were used as thresholds for interpreting the obtained results [44,45]. It is essential to assume that spices are not the only sources of these elements in food and therefore daily intake. However, we can estimate that the analyzed aromatic spices represent a valuable contribution, at least for Cu, to the everyday intake reported by consumers, with the highest concentrations found in fruits [*Prunus dulcis* (Mill.) D.A. Webb; *Pistacia vera* L.] and seeds (*Foeniculum vulgare* Mill.; *Papaver somniferum* L.) easily accessible in significant amounts. Therefore, according to the current results, the everyday consumption of certain elements can have adverse health effects through the repeated consumption of seeds (#4, #5, and #6) and fruits (#30). Of the 30 investigated samples, 27 and 29 spices exceeded the recommended limits (RDA) for Cu and Cr, respectively.

### 2.3. Potentially Toxic Trace Elements

As, Cd, Pb, and Hg are considered toxic trace elements, especially in food, due to their toxicity to humans. Lead and cadmium, for example, cause both acute and chronic poisoning and have adverse effects on the kidneys, liver, heart, and vascular and immune systems [46]. These elements are a subject of great concern, and their levels, especially in aromatic spices, need careful monitoring. The TE limits imposed by the European Pharmacopoeia [47] for herbal drugs are set only for Pb (5 mg/kg), Cd (1 mg/kg), and Hg (0.1 mg/kg) [48]. WHO recommendations for the maximum permissible levels of Cd and Pb in medicinal plants are 0.3 mg/kg and 10 mg/kg, respectively [49]. Even European legislation for food additives sets maximum limits only for Pb (0.3 mg/kg in leafy vegetables) and Cd (0.2 mg/kg in leafy vegetables and herbs) [13]. According to our measurements, the mean concentrations of Pb, Cd, As, and Hg were 0.21, 0.06, 0.10, and 0.07 μg/g, respectively (Figure 1). Compared with the standard limits, sample #14 had the highest content of Pb (1.42 μg/g). Other spices that exceeded the recommended FAO/WHO level for Pb [50] included #8, #11, #15, #17, #18, and #22 (1.18, 0.56, 0.48, 0.58, 0.30, and 0.31 μg/g, respectively). The concentrations of Cd in the majority of the samples under investigation were below the maximum limits permitted by European Community legislation in vegetable and condiment samples [13], with some exceptions, as in the case of #5 (0.22 μg/g) and #17 (0.29 μg/g). The high levels of cadmium may have originated from cadmium-containing phosphate fertilizers, the practice of growing plants in soil treated with sewage sludge, or a combination of both. Some samples showed no detectable amount of cadmium, such as #13, #16, #20, #25, and #26.

Arsenic values were significantly higher in #14 (0.42 μg/g), #30 (0.32 μg/g), #9 (0.29 μg/g), and #17 (0.22 μg/g) (Figure 1). In general, the TE content in spices reflects environmental pollution levels, bioaccumulation in plant tissue, and the application of trace elements containing materials such as arsenate-based pesticides [35]. According to PTWI levels for As, Pb, and Cd established at 15, 7, and 25 µg/kg bw, respectively, the results of the present survey pose no concern to humans (Table 2). The consumption of fish is by far the most significant source of ingestion-related mercury exposure in humans. From the literature, we can appreciate that plants and livestock are subjected to mercury bioaccumulation from soil, water, and the atmosphere due to the magnification of other mercury-containing resources [48]. The recorded range of Hg for the samples investigated (<0.001 to 0.534 ± 0.037 μg/g) was substantially below the WHO limit of 10 mg/kg [49]. Therefore, considering the level of Hg, we can state that the sampled spices are sufficiently safe.

### 2.4. Correlation Analysis

Pearson product-moment correlation analysis of the total elemental content for all spices in each category is presented in Figure 2, showing the relationships between elemental concentrations. Thus, significant positive correlations were noted for seeds at *p* < 0.5 between Co–As, Co–Pb, Cu–Cd, and Cu–Hg, with negative correlations between Co–Cd, Co–Hg, Ni–As, and Ni–Cu–Pb. Furthermore, a significant positive correlation was calculated for leaves between Co–As, Co–Hg, Ni–Pb, and Ni–As at a significance level of *p* < 0.01. At a significance of *p* < 0.5, the relationship was significant between Cr–Cd, Cr–Hg, Ni–Hg, Ni–Cd, Co–Pb, and Co–Cd. In addition, we observed a significant positive intermetal correlation between Cr–Hg, Cr–Pb, Co–As, Co–Hg, Ni–Hg, Ni–Pb, Cu–Cd, and Cu–Pb in the powdered spices. The results highlighted contrasting relationships for beans, noting positive and robust correlations of Cd versus Co and Ni (*p* < 0.01). At *p* < 0.5, the associations included positive coefficients (Cr–Cd, Cr–As, Co–As, Ni–As, and Ni–Cd) and negative coefficients for Cu versus Cd and As and Hg versus Cr, Co, and Ni. Finally, for fruits at *p* < 0.5, significant positive intermetal values were calculated for As versus Co and Cu and Cd–Ni, with negative values between As–Ni, Hg–Cr, and Cd versus Cr and Cu. A significant negative intermetal relationship revealed similar origins and positive conditioning between metals absorbed by plants [51]. We calculated a high correlation value (*p* < 0.001) for the case of Co–As when included in the analysis of all five datasets, which can be explained by the potential for cobalt-loaded resin to remove As(V).

Principal component analysis was applied to the total metal concentration data matrix to differentiate common metal patterns in all spices, using factor analysis by PCA extraction (Figure 3c). According to the Kaiser criterion, three principal components with eigenvalues of 2.52, 2.02, and 1.46 were extracted, comprising 75.21% of the total variance. An association between Pb, As, Co, and Ni in one group and Cr, Cu, Hg, and Cd in the second group was prominent, comprising 56.85% of the total variance. Cr, Hg, and Cd were characterized by positive factors (F1, F2, and F3) compared with Cu, which demonstrated positive F1 and F2 values (Figure 3a) and a negative F3 value (Figure 3b). As and Co were associated with positive loading F1 values and negative F2 and F3 values, respectively, while positive F1 and F3 and negative F2 values were highlighted for Ni and Pb, respectively. The highest positive loading values on F1 were calculated for #14 (2.26), #17 (1.82), and #6 (1.76), with negative values for #16 (−1.43) and #20 (−1.40). An extreme positive value was observed for F2 associated with #6 (4.41), and a negative value was observed for F3 associated with #30 (−3.93). The HCA highlighted which metals were associated in the spices in terms of their similarity and supported the PCA and correlation analysis (Figure 3d,e).

### 2.5. Exposure Assessment of Trace Elements and Health Risk Characterization

The estimated daily intake (EDI) values of the minor and potentially toxic trace elements in each spice were calculated, and the mean values followed the order Cu > Ni > Cr > Pb > Co > As > Hg > Cd (Figure 4). For all samples, the EDI values were considerably lower than the provisional tolerable daily intake, which indicates that consumption of the analyzed condiments does not pose a significant risk to consumer health. The highest values were calculated in #6 for Cd and Hg (0.09 and 0.07 µg/kg bw/day, respectively), #14 for Ni, As, and Pb (0.60, 0.05 and 0.20 µg/kg bw/day, respectively), #22 for Cr (0.63 µg/kg bw/day), and #30 for Co and Cu (0.18 and 3.42 µg/kg bw/day, respectively). The target hazard quotient (THQ) values followed the order Cr > As > Hg > Cu > Ni > Cd and Pb > Co (Figure 4).

All THQ levels and the total target hazard quotient (TTHQ) were below 1, indicating that there are no significant health risks associated with the intake of individual TE or mixtures through condiment ingestion. The highest THQs were recorded in #6, #14, #17, #22, and #30. Exposure to trace elements such as As, Hg, Cd, or Pb is frequently associated with various cancers (of the skin, lungs, or liver) or severe damage to the nervous system (as in the case of Pb). The cancer risk index was higher for Cd (3.2 × 10^−8^ − 3.7 × 10^−5^) than for As (1.6 × 10^−9^ − 2.5 × 10^−6^), Pb (3.7 × 10^−9^ − 1.7 × 10^−6^), and Hg (9.8 × 10^−12^ − 6.4 × 10^−10^). The total CR, calculated as the sum of the individual CRs for As, Pb, Cd, and Hg, ranged between 2.8 × 10^−8^ and 3.8 × 10^−5^ (Figure 4). For all elements investigated, the cancer risk index was below or within the safe limits of 10^−6^–10^−4^, indicating that lifetime exposure is negligible and consumption of the spices does not increase the risk of carcinogens.

## 3. Materials and Methods

### 3.1. Sample Preparation and Digestion

In June–September 2020, thirty food items of various origins were procured from several hypermarkets in Romania (Bucharest, Timisoara, and Cluj). For each foodstuff investigated, samples from five different lots were purchased from various stores to have representative samples as far as possible. The common, scientific, and family names of the plants are summarized in Table 3. The sampled technique was performed as replication. To determine the chemical composition of oils with significant organic matter content, total digestion of the matrix was mandatory to ensure complete metal dissolution. Therefore, the samples were oven-dried at 60 °C for ~12 h until a constant weight was obtained before chemical analysis. Then, the dried samples were ground in a stainless-steel mill until fine particles were obtained that could pass through a 0.5 mm mesh, which was kept dry for analysis. The contents of five different batches were mixed. First, 0.5 g of each sample was digested using the closed iPrep vessel speed iWave J MARS 6 CEM One Touch system (CEM Corporation, Matthews, North Carolina, USA). Then, we used a mixture of concentrated acids (10 mL of 69% HNO_3_) according to the two-step, temperature-controlled, Microwave Digestion of Pepper program (Bell, Chili, etc., CEM MARS 6 Method Note Compendium, 2019). After cooling, the contents of the tubes were transferred to a 50 mL volumetric flask with plug seal caps filled to volume with ultrapure water. Blank samples were also prepared by following the analytical methodology mentioned above. These solutions were analyzed by ICP-MS after appropriate dilution using external standards for calibration, considering five points on the curve and one for quality control.

### 3.2. Instrumentation

Multi-elemental analysis by ICP-MS started with the decomposition of the foodstuff samples. Quantitative determination of the elements Cr, Ni, Cu, As, Cd, Hg, Pb, and Co in spices was performed using an inductively coupled plasma mass spectrometer, ICP-MS Plasma Quant Elite (Analytik Jena, Germany), equipped with an AIM 3300 Autosampler (Analytik Jena, Germany) and collision-reaction interface iCRI working in H_2_ and He modes. The optimal conditions were as follows: RF power of 1.28 kW, plasma gas flow rate of 9 L/min, auxiliary gas flow rate of 1.5 L/min, nebulizer gas flow rate of 1.01 L/min, H_2_ gas flow rate of 90 mL/min, He gas flow of 120 mL/min, Ar gas flow of 10 mL/min, a dwell time of 50 ms for ^52^Cr, ^60^Ni, ^112^Cu, ^75^As, ^111^Cd, ^202^Hg, ^208^Pb, and ^59^Co, and ^209^Bi, ^6^Li, ^45^Sc, ^159^Tb, and ^89^Y isotopes that served as internal standards.

### 3.3. Chemical Reagents and Material Standards

Ultrapure nitric acid (HNO_3_, 60%) from Merck was used for sample digestion. Ultrapure deionized water with a resistivity of 18.2 MΩ·cm^−1^ was obtained from a Milli-Q water purification system (Millipore, Bedford, MA, USA). All the containers (polypropylene) were cleaned with 10% (*v*/*v*) nitric acid and deionized water. All plastic labware used for the sampling and sample treatment was new or cleaned by soaking first for 24 h in 10% (*v*/*v*) HNO_3_, then in ultrapure water. Calibration standard solutions and internal standards were prepared by successive dilution of a high purity ICP multi-element calibration standard (10 μg/L from twenty-nine elemental ICP-MS standards, matrix: 5% HNO_3_, Perkin Elmer Life and Analytical Sciences) and a mono-elemental calibration standard (10 μg/mL Hg, matrix 5% HNO_3_, Perkin Elmer Life and Analytical Sciences). We used three certified materials, NCS ZC85006—tomato (China National Analysis Center for Iron and Steel), IAEA-359—trace and minor elements in cabbage (IAEA Laboratory, Austria), and BOVM-1—bovine muscle certified reference material for trace metals and other constituents (National Research Council, Canada).

### 3.4. Quality Assurance

The analytical method used for assessing elements in spices was validated by measuring several quality parameters, such as sensitivity, linearity, precision, accuracy, and recovery. Linearity was established using calibration solution calibration curves. The sensitivity of the instrument was estimated through the determination of detection limits for all elements studied. The limit of detection (LOD) and limits of quantification (LOQ) were calculated by three and ten times the standard deviation of the blank sample divided by the slope of the analytical curve, respectively. The ICP-MS Plasma Quant Elite approach of three standard deviations was used to estimate the limits of detection. Using ultrapure water of 18.2 MΩ·cm^−1^, the signal intensities for the blank sample were recorded, and solutions of 10 μg/L were used for Cr, Cd, Ni, As, Co, Hg, Pb, and Cu. We evaluated the precision using the relative standard deviation of 10 repeated determinations of one sample to exclude sample variability. As a result, a percentage coefficient of variation (CV%) was obtained for all analyte elements. The validity of the developed method was confirmed by analyzing certified reference materials (CRMs); namely, NCS ZC85006 for trace elements in tomato samples, IAEA-359 for trace elements in cabbage samples, and BOVM-1 bovine muscle for trace elements in meat samples. These certified reference materials were also mineralized similarly to the samples. The measurement accuracy was determined by comparing the analyte concentrations with the certified values and was expressed as percentage recovery, R (%).

### 3.5. Statistical Analysis

Data processing and statistical analysis were conducted using the IBM SPSS Statistics 27 software. Mean values were subjected to principal component analysis (PCA), hierarchical cluster analysis (HCA), and correlation analysis. The estimated daily intake of trace elements through condiment ingestion was calculated using Equation (1), as proposed by the FAO/WHO (Food and Agriculture Organization of the United Nations/World Health Organization) [52].
EDI = (C × IR)/BW,(1)
where C is the concentration of metal in condiments (mg/kg), IR is the average intake rate of spices (0.01 kg/person/day), and BW is the average body weight of an individual (70 kg). The obtained results were compared with the provisional tolerable daily intake (PTDI) for food. Furthermore, the noncarcinogenic health risk associated with trace elements exposure was evaluated using the method proposed by the US-EPA [53] by calculating the target hazard quotient for each metal according to the following equation:THQ = (EF × ED × IR × C)/(BW × AT × RfD),(2)
where EF is the exposure frequency (365 days/year); ED is the exposure duration (75.88 years is the average lifetime for adults, according to the National Institute of Statistics); IR is the average intake rate of condiments (0.01 kg/person/day); C is the concentration of metal in condiments (mg/kg); BW is the average body weight of an individual (70 kg); AT is the average exposure time (365 days/year × 75.88 years); and RfD is the oral reference dose, which is 3 µg/kg bw/day (Cr), 43 µg/kg bw/day (Co), 20 µg/kg bw/day (Ni), 0.304 µg/kg bw/day (As), 1 µg/kg bw/day (Cd), 3.57 µg/kg bw/day (Pb), and 40 µg/kg bw/day (Cu). Considering the RfD level for methylmercury, in the present study, PTDI (0.571 µg/kg bw/day) was used as the RfD for Hg. If the THQ is below a value of 1, the risk of encountering adverse health effects is low, even for sensitive populations. However, if the THQ is equal to or greater than 1, health hazards may be associated with consuming contaminated condiments [53]. By summing the THQ for each trace element, the Total Target Hazard Quotient was obtained, which was assessed to evaluate the cumulative potential health risk caused by exposure to a mixture of trace elements. Furthermore, a TTHQ higher than 1 indicates a significant health concern due to the interactive effects of TE. The carcinogenic health risk associated with exposure to carcinogens such as Cd, Hg, and Pb was evaluated by calculating the cancer risk (CR) based on a method proposed by the US-EPA. The individual CR (for As, Cd, Hg and Pb) was calculated according to the following equation:CR = EDI × SF,(3)
where EDI is the estimated daily intake of the individual metal through condiments, and SF is the carcinogenic slope factor (upper bound approximating a 95% confidence limit). The SF values were 1.5 (As), 0.38 (Cd), 0.0085 × 10^−3^ (Hg), and 0.0085 (Pb). We evaluated the carcinogenic risk associated with exposure to a mixture of carcinogens by calculating the total carcinogenic risk (TCR) as the sum of the individual CRs for As, Cd, Hg, and Pb. The acceptable range for the TCR was within 10^−6^ to 10^−4^. For a TCR that exceeded the tolerable limit, the metals found in the condiments pose an unacceptable carcinogenic risk to human health. In the present study, it was assumed that inorganic arsenic constitutes 3% of the total arsenic.

## 4. Conclusions

The concentrations of four essential elements (Cr, Co, Ni, and Cu) and four toxic metals (Hg, Cd, Pb, and As) in 30 different spices collected from the Romanian market were investigated using microwave digestion and inductively coupled plasma-mass spectrometry. The raw materials originated from 16 countries, and nine samples across almost every category were local. Although our study supports the already published levels of trace elements in condiments and spices, the present study presented elemental variability between the origin of raw material, sources, and different assimilation pathways. Our results support the idea of accumulating contaminant levels in organisms after long-term ingestion. Thus, the Cu and Cr contents were higher than the RDA maximum permissible levels in 27 and 29 foodstuffs, respectively. According to TUL recommendations, even everyday consumption of elemental Ni and Cu in seeds (fennel, opium poppy, and cannabis) and fruits (almond) can have adverse health effects. A very high correlation coefficient was found between Cu and As, with a significance level of *p* < 0.001. Three principal components, comprising 75.21% of the total variance, were extracted using the Kaiser criterion. The first two groups associated Pb, As, Co, and Ni and Cr, Cu, Hg, and Cd, comprising 56.85% of the total variance. Cluster analysis classified the foodstuffs according to metal concentration similarity and demonstrated an association between the spice categories (leaves, fruits, seeds, powders, and beans) and similar metal uptake by the plant. The exposure assessment of TE and health risk characterization indicate no significant health impact on consumers after everyday consumption of the analyzed foodstuffs.

## Figures and Tables

**Figure 1 molecules-26-07081-f001:**
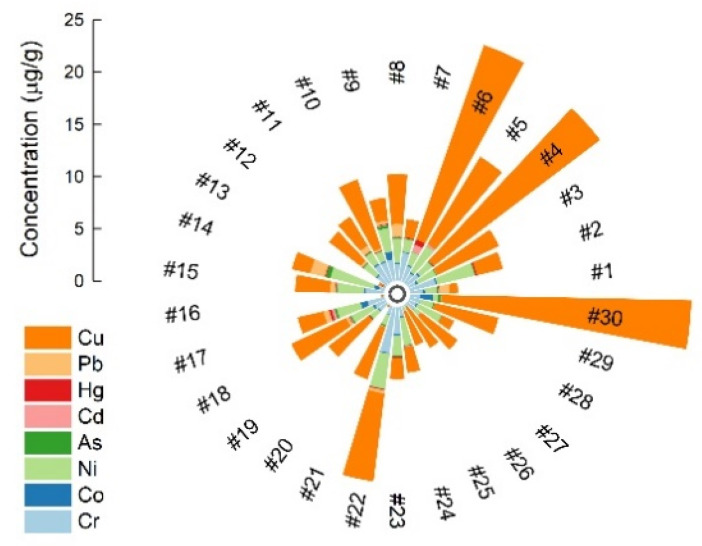
The eight elements metal content of thirtieth spices from Romanian market.

**Figure 2 molecules-26-07081-f002:**
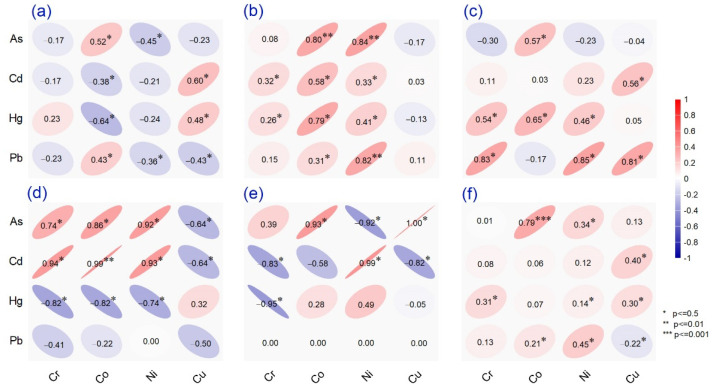
Pearson product-moment correlation for eight metals measured in thirtieth spices from Romanian market grouped as (**a**) seeds, (**b**) leaves, (**c**) powder, (**d**) beans, (**e**) fruits and (**f**) include all samples investigated.

**Figure 3 molecules-26-07081-f003:**
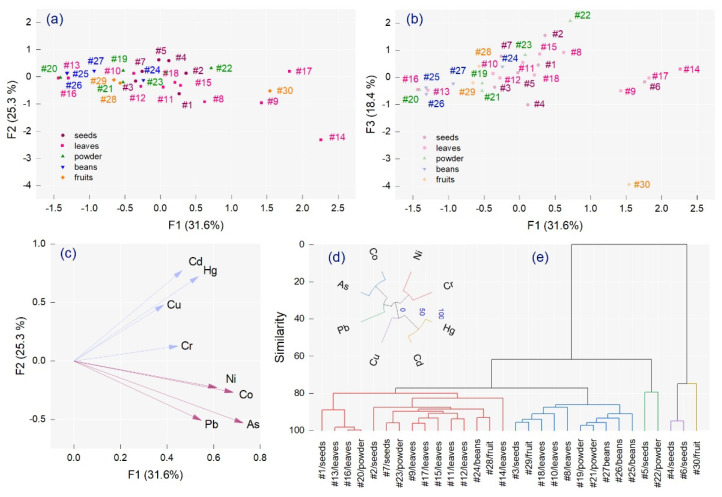
2D score plots for the 30 spices investigated from the Romanian market, (**a**) factor analysis (F1 versus F2), (**b**) factor analysis (F1 versus F3), (**c**) factor analysis loading plot by PCA extraction, (**d**) hierarchical cluster dendrogram for metals, and (**e**) hierarchical cluster dendrogram for samples.

**Figure 4 molecules-26-07081-f004:**
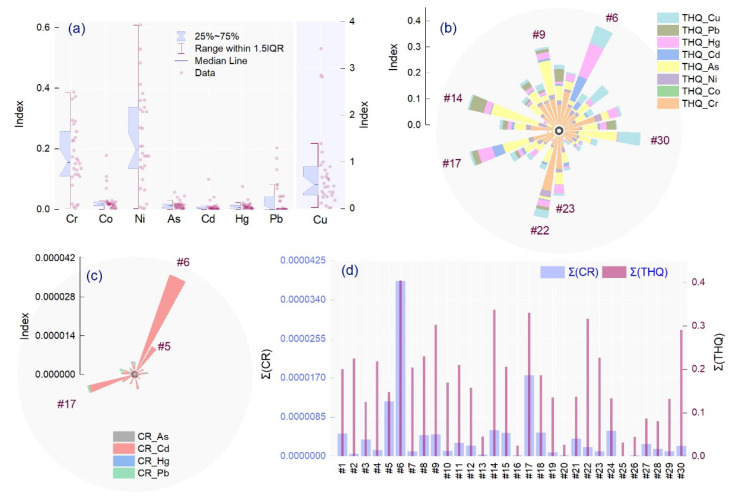
The graphical representation of (**a**) estimated daily intake (EDI), (**b**) Target Hazard Quotient (THQ), (**c**) Cancer Risk (CR), and (**d**) sum of Cancer Risk and Target Hazard Quotient.

**Table 1 molecules-26-07081-t001:** Analysis of certified reference material.

Element	*r^2^*	LOD(μg/g)	LOQ(μg/g)	RSD%	Analyzed Certified Reference Materials								
					CRM-NCS ZC85006			CRM-BOVM1			CRM-IAEA359		
					Certified Values ± sd	Measured Values ± sd	Recovery(%)	Certified Values ± sd	Measured Values ± sd	Recovery(%)	Certified Values ± sd	Measured Values ± sd	Recovery(%)
					Concentration (μg/g)			Concentration (μg/g)			Concentration (μg/g)		
Cu	0.9997	0.006	0.06	1.1	21.100 ± 2.5	19.080 ^a^ ± 0.2 ^b^	90	2.840 ± 0.450	2.694 ^a^ ± 0.034 ^b^	95	-	-	-
Cr	0.9998	0.001	0.01	7.2	-	-	-	0.071 ^c^	0.068 ± 0.004	96	1.300 ± 0.060	1.392 ^a^ ± 0.120 ^b^	107
Ni	0.9996	0.005	0.05	5.8	2.23 ^d^	2.345 ± 0.162	105	-	-	-	1.050 ± 0.050	0.947 ± 0.051	90
Co	0.9992	0.002	0.02	6.1	0.670 ± 0.040	0.600 ± 0.035	90	0.007 ± 0.003	0.006 ± 0.0004	86	-	-	-
As	0.9993	0.0005	0.005	2.1	1.050 ^d^	0.965 ± 0.015	92	-	-	-	0.10 ± 0.005	0.09 ± 0.005	90
Cd	0.9999	0.003	0.03	5.4	-	-	-	0.013 ± 0.011	0.012 ± 0.001	95	0.12 ± 0.006	0.116 ± 0.003	97
Pb	0.9995	0.0005	0.005	4.0	4.970 ± 0.540	4.688 ± 0.079	94	0.38 ± 0.24	0.343 ± 0.022	95	-	-	-
Hg	0.9990	0.0005	0.005	6.0	0.140 ± 0.015	0.117 ± 0.010	84	-	-	-	0.013 ± 0.0003	0.011 ± 0.0004	85

^a^ mean concentration of six replicate measurements; ^b^ standard deviation; ^c^ information value; ^d^ standard deviation is not available.

**Table 2 molecules-26-07081-t002:** Comparative reported values for the investigated elements in some widely used spices: this study versus other references.

Spices	Element Concentration in μg/g								References
	Cr	Co	Ni	Cu	Pb	As	Cd	Hg	
Thyme	1.664 ± 0.083	0.676 ± 0.040	2.375 ± 0.095	2.547 ± 0.102	0.583 ± 0.052	0.221 ± 0.013	0.286 ± 0.017	0.230 ± 0.016	This work
	1.610 ± 0.040	0.520 ± 0.030	6.140 ± 0.280	-	0.500 ± 0.030	0.310 ± 0.030	ND	-	[34]
	0.970 ± 0.110	0.193 ± 0.019	2.340 ± 0.090	12.170 ± 0.520	-	0.277 ± 0.005	0.026 ± 0.006	-	[37]
	3.660 ± 0.960	-	2.290 ± 0.550	1.900 ± 0.640	1.590 ± 0.380	-	0.760 ± 0.170	-	[38]
Fennel	1.367 ± 0.064	0.157 ± 0.009	1.459 ± 0.073	19.967 ± 0600	<0.001	0.109 ± 0.009	0.012 ± 0.001	0.076 ± 0.005	This work
	0.280 ± 0.010	0.150 ± 0.002	2.860 ± 0.020	5.270 ± 0.040	0.391 ± 0.008	0.067 ± 0.001	0.015 ± 0.004	-	[17]
	35.930 ± 9.120	-	28.660 ± 1.380	8.280 ± 0.850	0.350 ± 0.110	-	0.500 ± 0.110	-	[38]
	1.500 ± 0.350	0.540 ± 0.100	1.460 ± 0.034	10.900 ± 0.800	-	-	0.086 ± 0.034	-	[39]
Cinnamon	0.023 ± 0.008	0.035 ± 0.002	0.028 ± 0.002	0.116 ± 0.009	<0.001	0.030 ± 0.002	<0.001	0.040 ± 0.004	This work
	0.160 ± 0.020	0.070 ± 0.001	ND	-	0.070 ± 0.010	ND	0.210 ± 0.020	-	[34]
	0.420 ± 0.010	0.086 ± 0.0009	0.940 ± 0.010	2.830 ± 0.010	0.162 ± 0.015	0.056 ± 0.006	0.026 ± 0.001	-	[17]
	0.520 ± 0.040	0.060 ± 0.002	0.320 ± 0.050	2.780 ± 0.020	-	0.034 ± 0.004	0.118 ± 0.009	-	[37]
	2.210 ± 0.220	0.130 ± 0.001	0.720 ± 0.006	4.680 ± 0.290	0.310 ± 0.060	-	-	-	[40]
Cumin	2.552 ± 0.127	0.132 ± 0.011	3.71 5± 0.149	2.547 ± 0.51	<0.001	0.063 ± 0.005	0.001 ± 0.0009	0.154 ± 0.010	This work
	1.170 ± 0.120	0.260 ± 0.100	2.450 ± 0.050	-	0.140 ± 0.010	0.190 ± 0.020	0.050 ± 0.001	-	[34]
	0.380 ± 0.010	0.051 ± 0.004	1.590 ± 0.010	7.400 ± 0.030	0.258 ± 0.002	0.072 ± 0.001	0.032 ± 0.0005	-	[17]
	1.790 ± 0.210	0.234 ± 0.040	2.560 ± 0.090	8.330 ± 0.540	-	0.174 ± 0.014	0.045 ± 0.009	-	[37]
	24.860 ± 4.300	-	27.04 ± 1.670	2.780 ± 0.140	0.890 ± 0.280	-	0.590 ± 0.110	-	[38]
Black pepper	0.113 ± 0.005	0.003 ± 0.0002	0.123 ± 0.009	4.506 ± 0.315	<0.001	0.027 ± 0.002	<0.001	0.035 ± 0.003	This work
	0.580 ± 0.050	0.140 ± 0.001	1.630 ± 0.010	-	ND	0.020 ± 0.010	ND	-	[34]
	0.710 ± 0.050	0.131 ± 0.003	4.040 ± 0.600	10.190 ± 0.880	-	0.028 ± 0.003	<LOD	-	[37]
	-	-	-	2.400 ± 0.100	-	-	-	-	[41]
	5.270 ± 0.240	-	4.740 ± 1.170	4.780±0.590	0.880 ± 0.140	-	0.790 ± 0.300	-	[38]
Ginger	1.974 ± 0.108	0.128 ± 0.007	0.984 ± 0.039	6.995 ± 0.461	0.192 ± 0.011	0.048 ± 0.004	0.011 ± 0.001	0.049 ± 0.0005	This work
	0.800 ± 0.120	0.410 ± 0.020	1.140 ± 0.130	-	0.370 ± 0.001	0.030 ± 0.001	0.070 ± 0.001	-	[34]
	-	-	-	1.600 ± 0.100	-	-	-	-	[41]
	3.060 ± 0.001	0.400 ± 0.001	2.260 ± 0.120	17.600 ± 2.700	3.010 ± 0.600	-	-	-	[40]
Coriander	0.912 ± 0.045	0.120 ± 0.007	2.685 ± 0.107	2.445 ± 0.122	0.003 ± 0.0002	0.088 ± 0.007	0.091 ± 0.007	0.031 ± 0.002	This work
	<0.08	<0.08	2.20 (RSD < 2%)	5.500 (RSD < 2%)	<0.010	<0.010	<0.010		[42]
	0.410 ± 0.010	0.131 ± 0.009	1.350 ± 0.020	7.820 ± 0.040	0.213 ± 0.045	0.028 ± 0.0006	0.026 ± 0.001	-	[17]
	-	-	-	-	0.150 ± 0.005	0.031 ± 0.002	0.084 ± 0.005	0.001 ± 0.0002	[43]

**Table 3 molecules-26-07081-t003:** List of aromatic spices studied.

No	Botanical Name	English Name	Part Investigated	Origin
#1	*Pimpinella anisum* L.	Anise	seeds	Romania
#2	*Carum carvi* L.	Caraway	seeds	Finland
#3	*Anethum graveolens* L.	Dill	seeds	Romania
#4	*Foeniculum vulgare* Mill.	Fennel	seeds	Romania
#5	*Linum usitatissimum* L.	Flaxseed	seeds	Ukraine
#6	*Papaver somniferum* L.	Opium poppy	seeds	Romania
#7	*Cannabis sativa* L.	Cannabis	seeds	China
#8	*Allium ursinum* L.	Ramsons	leaves	Romania
#9	*Ocimum basillicum* L.	Basil	leaves	Egypt
#10	*Zingiber officinale* Roscoe	Ginger	leaves	Nigeria
#11	*Laurus nobilis* L.	Bay laurel	leaves	Romania
#12	*Levisticum officinale* W.D.J. Koch	Lovage	leaves	Poland
#13	*Melissa officinalis* L.	Lemon balm	leaves	Romania
#14	*Origanum vulgare* L.	Oregano	leaves	Turkey
#15	*Petroselinum crispum* (Mill.) Fuss	Parsley	leaves	Poland
#16	*Salvia rosmarinus* Spenn.	Rosemary	leaves	Romania
#17	*Satureja hortensis* L.	Summer savory	leaves	Germany
#18	*Thymus serpyllum* L.	Thyme	leaves	Romania
#19	*Pimenta dioica* L. Merr.	Allspice	powder	Bulgaria
#20	*Cinnamomum verum* J. Presl	Cinnamon	powder	India
#21	*Syzygium aromaticum* (L.) Merr. & L.M. Perry	Clove	powder	India
#22	*Myristica fragrans* Houtt.	Nutmeg	powder	Indonesia
#23	*Capsicum annuum* L.	Paprika	powder	China
#24	*Coriandrum sativum* L.	Coriander	beans	Bulgaria
#25	*Piper nigrum* L.	Green pepper	beans	Austria
#26	*Piper nigrum* L.	Black pepper	beans	Vietnam
#27	*Sinapis alba* L.	White mustard	beans	India
#28	*Juniperus communis* L.	Common juniper	fruit	Ukraine
#29	*Pistacia vera* L.	Pistachio	fruit	Greece
#30	*Prunus dulcis* (Mill.) D.A. Webb	Almond	fruit	Australia

## Data Availability

All data relevant to the study are included in the article.

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
