# Peer review of "ICP-MS Assessment of Essential and Toxic Trace Elements in Foodstuffs with Different Geographic Origins Available in Romanian Supermarkets"

_molecules, 2021, doi:10.3390/molecules26237081_

Round 1

Reviewer 1 Report

The authors describe the results of monitoring essential elements (Cr, Co, Ni, Cu) and toxic metals (Hg, Cd, Pb, and As)  toxicity of frequently consumed food spices and provide important data on herbs produced in Romania. The results are interesting because analyze the toxic side effects of 
contamination in the context of nationally and internationally imposed limits. The authors explain that exposure assessment  and health risk characterization indicate no significant health impact on consumers after everyday consumption of the analyzed foodstuffs. I think that this conclusion needs to be better supported,  paper focuses too much on validation and little on food safety aspects of food. The manuscript can be improved with some items indicated below. Paper below could be cited to improve the introduction:

  1. Outbreak of fatal nitrate toxicosis associated with consumption of fennels (Foeniculum vulgare) in cattle farmed in Campania region (southern Italy) A Costagliola, F Roperto, D Benedetto, A Anastasio, R Marrone, A Perillo, ... Environmental Science and Pollution Research 21 (9), 6252-6257

Explain better foods sampling and selection procedure.

Check English and References section.

Author Response

Dear editor,

We address sincere gratitude for a careful and thorough reading of the manuscript and the thoughtful comments and constructive suggestions, which helped improve the quality of the draft. According to the recommendations, we have close revised our manuscript, and its final version is enclosed. Point-by-point responses to the comments are listed below. We hope that the revised manuscript is acceptable for publishing. In addition, a double-check of English grammar using Springer Nature Author Services and references was achieved.

The authors describe the results of monitoring essential elements (Cr, Co, Ni, Cu) and toxic metals (Hg, Cd, Pb, and As) toxicity of frequently consumed food spices and provide important data on herbs produced in Romania. The results are interesting because analyze the toxic side effects of contamination in the context of nationally and internationally imposed limits. The authors explain that exposure assessment and health risk characterization indicate no significant health impact on consumers after everyday consumption of the analyzed foodstuffs. I think that this conclusion needs to be better supported, paper focuses too much on validation and little on food safety aspects of food. The manuscript can be improved with some items indicated below. Paper below could be cited to improve the introduction:

  1. Outbreak of fatal nitrate toxicosis associated with consumption of fennels (Foeniculum vulgare) in cattle farmed in Campania region (southern Italy) A Costagliola, F Roperto, D Benedetto, A Anastasio, R Marrone, A Perillo, ... Environmental Science and Pollution Research 21 (9), 6252-6257

Response: We included in the Introduction chapter a paragraph discussing other challenges in nutritional foodstuff as is the case of nitrite and nitrate, as follows: "Contamination with TCE is by far not the only issue in consuming vegetables. The contamination of soil, groundwater, and surface water with nitrate and nitrite is responsible for plant enrichment with nitrate and nitrite. Due to its natural form and additive presence, a new challenge in the risk-benefit health assessment of vegetable food enriched with nitrate is frequently discussed. In agricultural practices, to prevent the growth of the bacterium Clostridium botulinum, impressive amounts of nitrate and nitrite are also used to promote some food colors. Nitrate-accumulating vegetable content varies between species and genotypes with different ploidies, and high contents have been mentioned for Brassicaceae, Chenopodiaceae, Amaranthaceae, and Apiaceae. Past studies have indicated that contaminated fennel led to nitrate toxicosis that was associated with cattle side effects (muscular tremors, respiratory distress, and convulsions) [56]. A carcinogenic risk is related to the formation of nitrosamine compounds."

Explain better foods sampling and selection procedure.

Response: The Material and Method chapter was double-checked, and detailed information was added for the sampling selection procedure.

Check English and References section.

Response: The manuscript was corrected for English flow, correct expression and consistency by Nature Editing Services. Verification Code: D2BC-10A5-58F9-C239-97BB.

Reviewer 2 Report

The manuscript gives interesting results on the essential and non essential metals in different typologies of spices and other foodstuffs. The novelty of the present work relies on the contribution of new data regarding the presence of toxic metals in spices not reported before. However, the paper could contribute to the scientific community after a minor revision. All my suggestions and questions are listed below:

TITLE

The AA should replace the term heavy metals with trace elements.

INTRODUCTION

Line 91-95: this part is suitable for the material and methods section

MATERIALS AND METHODS

Lines 289-308: All these information can be reassumed in a table

Lines 310-313: This part is unnecessary for the materials and methods of this work

Line 314: “cleaned”. How they were cleaned?

Line 322: Did the samples were made up to volume with ultrapure water after the digestion? Please specify

Line 325: How many calibration points were considered?

Line 328-333: The AA reported again the digestion protocol here.

Line 362: How many concentration points were considered for the linearity determination?

Line 365: “Treshold” do the AA mean Limit?

CONCLUSIONS

Line 425-430: This part is suitable for the abstract, not for the conclusions.

Author Response

Dear editor,

We address sincere gratitude for a careful and thorough reading of the manuscript and the thoughtful comments and constructive suggestions, which helped improve the quality of the draft. According to the recommendations, we have close revised our manuscript, and its final version is enclosed. Point-by-point responses to the comments are listed below. We hope that the revised manuscript is acceptable for publishing. In addition, a double-check of English grammar using Springer Nature Author Services and references was achieved.

The manuscript gives interesting results on the essential and non essential metals in different typologies of spices and other foodstuffs. The novelty of the present work relies on the contribution of new data regarding the presence of toxic metals in spices not reported before. However, the paper could contribute to the scientific community after a minor revision. All my suggestions and questions are listed below:

TITLE

The AA should replace the term heavy metals with trace elements.

INTRODUCTION

Line 91-95: this part is suitable for the material and methods section

Response: We corrected the sentence, and the flow and clarity of the sentence were improved as follows "Therefore, the objective of the present study is (i) to quantify the levels of essential trace elements (Cr, Co, Ni, and Cu) and toxic trace elements (Hg, Cd, Pb, and As) in the food items widely used and commonly available on the market and consumed in the main cities of Romania, (ii) to establish the relationship between trace elements in foodstuff, and (iii) to assess the contribution of essential and toxic trace elements to human nutritional assimilation and the health risk due to the ingestion according to limitation established by WHO/FAO regulation."

MATERIALS AND METHODS

Lines 289-308: All these information can be reassumed in a table.

Response: We added Table 3, in which we summarized all information regarding the sample investigated.

Lines 310-313: This part is unnecessary for the materials and methods of this work

Response: The paragraph was deleted. Thank you for your recommendation.

Line 314: "cleaned". How they were cleaned?

Response: The sentence was rephrased. Thank you!

Line 322: Did the samples were made up to volume with ultrapure water after the digestion? Please specify

Response: We added information regarding the digestion procedure.

Line 325: How many calibration points were considered?

Response: We rephrased as follows:" These solutions were analyzed by ICP–MS after appropriate dilution using external standards for calibration, considering five points on the curve and one for quality control."

Line 328-333: The AA reported again the digestion protocol here.

Response: We double-check the whole document for repeated sentences.

Line 362: How many concentration points were considered for the linearity determination?

Response: The information required was added to the sample preparation and digestion chapter. Thank you.

Line 365: "Treshold" do the AA mean Limit?

 Response: We corrected the sentence.

CONCLUSIONS

Line 425-430: This part is suitable for the abstract, not for the conclusions.

Response: We eliminated the sentence. Thank you for the suggestion.

Reviewer 3 Report

The manuscript is interesting but reference documents used for health risk estimation are very old, also old  reference values were used. In my opinion authors should base on novel references and threshold values and recalculated their results concerning with risk assessment.

Abbreviations should be written as they appear in the manuscript for the first time.

In abstract authors should provide explanation for HM abbreviation.

Sentence “In the samples analyzed, we noted no detectable side effects of toxic HMs” is unclear and should be expanded (what side effects authors mean?)

Line 58-59 Sentence is unclear “and is a natural proxy for natural and anthropogenic activities”.

Line 78 Give explanation of abbreviations ICP-AES and ICO-OES. There should be ICP-OES instead of ICO-OES.

Line 82 Give abbreviations for other techniques, for example (LA-ICP-MS-TOF) for laser ablation–inductively coupled plasma–time of flight mass spectrometry.

Line 133 "We measured extreme values of Ni....." authors should provide values for Ni concentration obtained by other authors.

Line 142 There is "include inducing...."

Line 172-176 Why authors use such old refernces (2001 and 1983) with reference values? Authors should take values established by EU and EFSA.

Why authors use so old documents and reference values  for risk assessments? Risk assessment should be recalculated base on novel documents.

Line  266-276  - provide full names of EDI, THQ and TTHQ.

Why author describe sample preparation in 3.1 and in 3.2 sections? Some information is written twice.

Author Response

Dear reviewer,

We address sincere gratitude for a careful and thorough reading of the manuscript and the thoughtful comments and constructive suggestions, which helped improve the quality of the draft. According to the recommendations, we have close revised our manuscript, and its final version is enclosed. Point-by-point responses to the comments are listed below. We hope that the revised manuscript is acceptable for publishing. In addition, a double-check of English grammar using Springer Nature Author Services and references was achieved.

The manuscript is interesting but reference documents used for health risk estimation are very old, also old  reference values were used. In my opinion authors should base on novel references and threshold values and recalculated their results concerning with risk assessment.

Abbreviations should be written as they appear in the manuscript for the first time.

Response: The document was double-checking for abbreviation usage and consistency. Thank you for recommendation.

In abstract authors should provide explanation for HM abbreviation.

Response: The heavy metal was eliminated from the manuscript following reviewer #2 recommendation, and the whole manuscript was revised accordingly. Thank you.

Sentence "In the samples analyzed, we noted no detectable side effects of toxic HMs" is unclear and should be expanded (what side effects authors mean?)

Response: The sentence summarized the previous results mentioned and was deleted.

Line 58-59 Sentence is unclear "and is a natural proxy for natural and anthropogenic activities".

Response: The sentence was rephrased. Thank you for your recommendation.

Line 78 Give explanation of abbreviations ICP-AES and ICO-OES. There should be ICP-OES instead of ICO-OES.

Response: The abbreviation was inserted in the manuscript text.

Line 82 Give abbreviations for other techniques, for example (LA-ICP-MS-TOF) for laser ablation–inductively coupled plasma–time of flight mass spectrometry.

Response: The abbreviation was added to the main text. Thank you for the suggestion.

Line 133 "We measured extreme values of Ni....." authors should provide values for Ni concentration obtained by other authors.

Response: The sentence was rephrased.

Line 142 There is "include inducing...."

Response: The manuscript was corrected for English flow, correct expression and consistency by Nature Editing Services. Verification Code: D2BC-10A5-58F9-C239-97BB.

Line 172-176 Why authors use such old refernces (2001 and 1983) with reference values? Authors should take values established by EU and EFSA.

Response: All references was updated.

Why authors use so old documents and reference values for risk assessments? Risk assessment should be recalculated base on novel documents.

Response: We included new references in all manuscript.

Line 266-276 - provide full names of EDI, THQ and TTHQ.

Response: The abbreviation was explained in the text. Thank you for your recommendation.

Why author describe sample preparation in 3.1 and in 3.2 sections? Some information is written twice.

Response: The repeated method information was reconsidered. Thank you for your close suggestion.

Round 2

Reviewer 3 Report

Manuscript was corrected according to reviewers' commments.